# Natural Multi-Enriched Eggs with n-3 Polyunsaturated Fatty Acids, Selenium, Vitamin E, and Lutein

**DOI:** 10.3390/ani13020321

**Published:** 2023-01-16

**Authors:** Zlata Kralik, Gordana Kralik, Manuela Košević, Olivera Galović, Mirela Samardžić

**Affiliations:** 1Faculty of Agrobiotechnical Sciences Osijek, Josip Juraj Strossmayer University of Osijek, V. Preloga 1, 31000 Osijek, Croatia; 2Scientific Center of Excellence for Personalized Health Care, Josip Juraj Strossmayer University of Osijek, Trg sv. Trojstva 3, 31000 Osijek, Croatia; 3Nutricin j.d.o.o., Đure Đakovića 6, 31326 Darda, Croatia; 4Department of Chemistry, Josip Juraj Strossmayer University of Osijek, Cara Hadrijana 8a, 31000 Osijek, Croatia

**Keywords:** enriched eggs, n-3 fatty acids, selenium, vitamin E, lutein

## Abstract

**Simple Summary:**

Eggs are considered a complete animal food. They are rich in proteins, essential fatty acids, minerals, and vitamins. By modifying the composition of feed mixtures for laying hens, it is possible to influence the content of certain nutrients in eggs. By adding vegetable oils (linseed and/or rapeseed) to feed mixtures for layers, it is possible to increase the content of omega-3 fatty acids in eggs, more specifically, the content of α-linolenic fatty acid increases. However, for human health, essential fatty acids such as eicosapentaenoic acid (EPA) and docosahexaenoic acid (DHA) are important, and their content in eggs can be increased by adding fish oil to feed mixtures for hens. Given that omega fatty acids are subject to oxidation, it is desirable to design the mixtures for laying hens in such a way that the content of some antioxidants is also increased. The most common antioxidants used in egg enrichment are selenium, vitamin E, and lutein. Eggs enriched with the mentioned components are desirable in the human diet because they have a positive effect on human health and can be called functional food.

**Abstract:**

The research investigates the possibilities of enriching eggs with n-3 polyunsaturated fatty acids, selenium, vitamin E, and lutein. The research was carried out on 100 TETRA SL laying hens divided into two groups (C and E). Hens in treatment C (control group) were fed a standard feeding mixture that contained 5% soybean oil, 0.32 mg/kg organic selenium, 25.20 mg/kg vitamin E, and 20.5 mg/kg lutein (Marigold flower extract). Hens in treatment E (experimental group) were fed a mixture in which soybean oil was replaced by 1.5% fish oil + 1.5% rapeseed oil + 2.0% linseed oil. The content of other nutricines amounted to 0.47 mg/kg organic selenium, 125.2 mg/kg vitamin E, and 120.5 mg/kg lutein. Portions of total n-3 polyunsaturated fatty acids in enriched eggs were significantly increased when compared to conventional eggs (480.65:204.58 mg/100 g; *p* < 0.001). The content of selenium, vitamin E, and lutein was significantly higher (*p* < 0.001) in enriched eggs than in conventional eggs. The content of selenium in egg albumen was increased by 1.81 times, and in the yolk, it was increased by 1.18 times. At the same time, the content of vitamin E was 2.74 times higher, and lutein was 8.94 times higher in enriched eggs than in conventional eggs.

## 1. Introduction

Poultry production is one of the fastest growing animal industries globally. Given that there is a constant increase in the population on earth, and it is predicted that by 2050 the number of people will reach 9 billion, it is important to develop all segments of food production [1]. In the Republic of Croatia, poultry production was on the rise from 2013 to 2020. The number of poultry in 2021 was slightly reduced compared to the year before, amounting to 12,096,168 pcs. Out of the total number of poultry, the largest number are broilers (8,012,409 pcs.), followed by hens (3,616,598 pcs.). Chickens are raised on farms (3,229,332 pcs) and on private family farms (1,308,053 pcs). The total production of eggs in 2021 was 736,217,000 pieces. The average production of eggs per hen on farms is 228 pieces and on private family farms 150 pieces [2]. Eggs are a foodstuff of animal origin rich in nutrients, such as protein, fat, and vitamins. Being essential both for humans and animals, those nutrients have to be taken in through food. Many studies into the enrichment of eggs with functional ingredients, i.e., with nutricines, contribute to raising awareness on the necessity of their increased intake in the organism and the importance of assuring the “original” quality of eggs [3,4,5,6]. Daily human needs for nutrients that are increased in eggs in this paper are: 55 µg of selenium, 15 mg of vitamin E, and 10 mg of lutein (RDA). The recommended daily intake of DHA for infants is 100 mg, for pregnant and lactating women 200 mg [7], EPA and DHA 250 mg, ALA 1.6 g for men, and 1.1 g for women [8]. Eggs are suitable for modification of the fatty acid profile (FA), since the content of n-3 polyunsaturated fatty acids (n-3 PUFA), such as α-linolenic (ALA), eicosapentaenoic (EPA) and docosahexaenoic (DHA) acids can be increased. Regulation of the European Union No. 116/2010 [9] states that food is a source of omega-3 fatty acids if it contains at least 0.3 g ALA per 100 g and/or at least 40 mg EPA + DHA per 100 g. Food is a high source of omega-3 fatty acids if it contains at least 0.6 g ALA per 100 g and/or at least 80 mg EPA + DHA per 100 g. Eggs are also suitable for the deposition of higher concentrations of antioxidants (selenium, vitamin E, and lutein; [10]). Kralik et al. [6] with the addition of different proportions of fish oil to the mixtures for laying hens successfully increased the content of omega-3 fatty acids in eggs from 204.59 mg/100 g of eggs (control group without fish oil in the mixture) to 327.35 mg/100 g of eggs (group with the addition of 1.5% fish oil). Feng et al. [11] investigated the effect of the addition of microalgae oil or fish oil to hen feed on the enrichment of eggs with omega-3 fatty acids and gave a special review of the sensory quality of enriched eggs. The authors point out that the enrichment of total n-3 PUFA using fish oil is more successful than microalgae oil in mixtures for hens. Gajčević et al. [12] increased the content of selenium in egg yolks and egg whites by adding organic selenium to feed for laying hens (E1 0.2 mg/kg and E2 0.4 mg/kg). The content of selenium in egg whites increased from 231.5 ng/g to 345.0 ng/g and from 584.8 ng/g to 779.5 ng/g in egg yolks. The addition of marigold extract as a source of lutein to the feed of laying hens in an amount of 2% successfully increased the lutein content in egg yolks from 0.51 to 1.71 mg/kg [13]. Eggs enriched with nutricines have higher nutritive value and contain antioxidants that contribute to the maintenance of egg quality over a longer period of time [1]. Long-chain n-3 PUFAs are essential for humans. For a well-balanced diet, nutritionists recommend the ratio of n-6/n-3 PUFA to be 4-5:1. In order to achieve such a ratio in egg yolk lipids, it is necessary to modify feeding treatments with ingredients that can provide for such a ratio. Therefore, feeding mixtures for laying hens are supplemented with sunflower, fish, rapeseed, and linseed oil [6,14]. Fish oil is added to the feed of laying hens in limited quantities due to the possibility of an undesirable odor in the eggs [15]. Full-fat sunflower or flaxseed seeds, extruded linseed [16,17], and microalgae [18] or some commercial preparations are also added to the mixture for the purpose of enriching eggs with n-3 PUFAs and achieving their favorable ratio. Eggs enriched with n-3 PUFA are beneficial to the cardiovascular system and contribute to the prevention of diabetes, inflammation, and other diseases [19]. As a microelement with antioxidative action, selenium is important in the protection of lipids [20], proteins, and vitamins [21] against oxidation. Surai et al. [22] and Surai and Kochish [23] reported that dietary supplementation of selenium, vitamin E, and carotenoids affected the antioxidative status of poultry. The aim of the research was to enrich eggs with nutricines such as n-3 polyunsaturated fatty acids, selenium, vitamin E, and lutein, with the hypothesis that the nutrients added to feed are transferred to the eggs through the metabolism of the laying hens. To our knowledge, there is little information in the literature regarding the simultaneous enrichment of eggs with omega-3 fatty acids, selenium, vitamin E, and lutein. This research is complex because, in addition to determining the content of certain nutrients in enriched eggs compared to conventional eggs, the quality of fresh eggs and eggs stored in the refrigerator at +4 °C is also investigated, as well as the lipid oxidative process during egg storage. Consumption of enriched eggs contributes to human health, and the antioxidants contained therein maintain the quality of eggs during the storage period.

## 2. Materials and Methods

### 2.1. Housing and Feeding of Laying Hens

The research was carried out on 100 laying hens of the TETRA SL hybrid which were aged 32 weeks. Laying hens were divided into two groups: treatment C and treatment E, each with 50 laying hens distributed into five repetitions. Feeding treatment C contained 5% soybean oil, 0.32 mg/kg organic selenium, 25.2 mg/kg vitamin E, and 20.5 mg/kg lutein. In feeding treatment E 5% soybean oil was replaced with 1.5% fish oil + 1.5% rapeseed oil +2% linseed oil and content of nutricines amounted to 0.47 mg/kg organic selenium, 125.2 mg/kg vitamin E, and 120.5 mg/kg lutein. Experimental period lasted for 5 weeks. Laying hens were given feed and water ad libitum. Composition of feeding mixture and its chemical analysis are overviewed in Table 1. Both groups of laying hens were kept in enriched cages in the same poultry house with the same microclimatic conditions. Lighting regime applied during the experiment was 16 h of light and 8 h of darkness. Both C and E treatments were fed mixture balanced at 16.63% crude protein and 11.60 MJ/kg ME. Mixtures differed in the type and amount of supplemented oils, as well as in the content of selenium (Sel-Plex^®^, Alltech, Nicholasville, KY, USA), vitamin E (DSM, Heerlen, The Netherlands), and lutein Marigold flower extract (Phyto Nutraceutical Inc., Changsha, China).

### 2.2. Analysis of Egg Quality

Analyses into egg quality were performed on 100 eggs (50 per group) which were collected for three consecutive days at the end of the feeding trial. Eggs were analyzed both fresh and stored for 28 days in refrigerator at 4 °C. The following parameters of egg quality were examined: weight of whole egg and weight of main parts, portion of main parts in egg, shell strength and thickness, yolk color, pH of albumen, and pH of yolk. Main parts (albumen, yolk, and shell) were weighed by the PB 1502-S scale (Mettler Toledo, BBK 422-6 DXS), and portions of main parts were calculated (%). The following indicators of egg quality: egg weight (g), shell strength (kgf) and thickness (mm), albumen height (mm), Haugh units (HU), and yolk color were measured by an automatic device Digital Egg Tester—ET 6500 (Nabel Co., Ltd., Kyoto, Japan). The pH meter MP 120 (Mettler Toledo, model SevenEasy) was used for measuring of pH of albumen and yolk.

### 2.3. Chemical Analysis of Fatty Acid Profile

The profile of fatty acids was determined on 20 eggs (10 yolks from each group) and on 2 samples of feeding mixture (three parallel samples per treatment). The samples used for the analysis of the fatty acid profile in egg yolks and in feeding mixtures were prepared in a CEM MARS6 microwave. The fatty acid profile was determined in a gas chromatograph equipped with the flame ionization detector and the FAMEWAX capillary column (RESTEK, Bellefonte, PA, USA) (internal diameter 30 × 0.32 mm, film 0.25 µm).

### 2.4. Chemical Analysis of Selenium

The content of selenium in eggs was determined on 20 eggs from which we formed 10 yolk and 10 albumen samples per group. Content of selenium in feeding mixture was also analyzed. Feed and egg samples for selenium determination were prepared by digestion with HNO_3_ and H_2_O_2_ in a microwave oven (CEM, model Mars 6) for 25 min. After digestion, HCl was added to the sample. Prepared samples were dried at 90 °C and then cooled to room temperature. Selenium content in the samples was read on a Perkin Elmer Optima 2100 DV device (PerkinElmer Inc., Waltham, MA, USA) [24].

### 2.5. Chemical Analysis of Vitamin E

Chemical analysis of vitamin E contained in yolks was carried out on 10 eggs, i.e., on 5 yolks from each group. Content of vitamin E was also determined in the feeding mixture sample. There was 5 g of the sample weighed, transferred to a 50 mL centrifuge tube, and mixed with 15 mL methanol. The tube was vortexed and left in a dark place at room temperature for 16 h. After resting, the sample liquid was poured into the glass container and left again to settle down. Further analysis proceeded with supernatant filtered through a PTFE 0.20 μm microfilter into a container. The sample was then analyzed by HPLC system Shimadzu with UV–VIS detector equipped with Shim-pack GIST C-18 (150 mm × 4.6 mm, 5 μm) column. Measurement wavelength was 295 nm (UV area), mobile phase: propan-2-ol: methanol (45:55, *v*/*v*) (HPLC purity), flow rate 0.7 mL/min, retention time for tocopherol 4.7 min.

### 2.6. Chemical Analysis of Lutein

The content of lutein was determined on the sample of 20 eggs, i.e., 10 yolks per each group, as well as on the feeding mixture sample. Determination of lutein content in egg yolks and in feeding mixture was carried out according to the method of Leeson and Caston [25]. The sample was analyzed in the Shimadzu HPLC device using Shim-pack GIST C-18 (250 mm × 4.6 mm, 5 μm) column. The mobile phase was methanol and tetrahydrofuran (THF) 9:1 (*v*/*v*). The volume of injected sample was 20 µL. The flow rate was 1 mL/min, wavelength was 450 nm and the analysis lasted for 20 min. The standard curve of lutein was prepared by using a lutein standard purchased from ChromaDex (Irvine, CA, USA).

### 2.7. Determination of Lipid Oxidation in Egg Yolks

Lipid oxidation was determined on a sample of 20 yolks (10 yolks from each group). The sample was prepared and analyzed as follows: the yolk was weighed in a test tube and mixed with 10% trichloroacetic acid. The sample was then homogenized and centrifuged at 5500× *g* for 15 min at 4 °C. After centrifugation, supernatant was mixed with thiobarbituric acid (pH 2.5). Tubes were immersed in a water bath at 95 °C for 30 min. After cooling, distilled water was added and the sample was centrifuged at 5500× *g* for 15 min at 4 °C. Content of the colored product was measured spectrophotometrically at 532 nm. Results are compared with the standard curve of malondialdehyde tetrabutylammonium salt standard (Sigma-Aldrich, Buchs, Switzerland), and presented in µg MDA/g yolk.

### 2.8. Statistical Data Analysis

Data collected within egg analyses were statistically processed in the Statistica software [26] and overviewed in tables. Presented statistical parameters referred to arithmetic mean ($\bar{x}$) and standard deviation (sd). Testing the significance of differences between the groups was performed by the one-way and two-way analysis of variance (ANOVA). The calculated F value was compared with the theoretical F value at the significance level of 5%. Significance of differences between the mean values was determined by Fisher’s LSD test.

## 3. Results

Table 2 presents the fatty acid profile of feeding mixtures of treatments C and E. The content of fatty acids in feeding mixtures differed significantly between treatment C and treatment E. Referring to the research objective of enriching eggs with ALA, EPA, and DHA, i.e., with ∑n-3 PUFA, the feeding mixture of the treatment E contained more fatty acids than the treatment C (ALA 17.39%:4.24%, EPA 1.62%:0.49%, DHA 2.81%:0.00%, and ∑n-3 PUFA 21.83%:4.73%). The ratio of n-6/n-3 PUFA in feeding treatment E was 1.06:1, while in feeding treatment C, it was 9.32:1.

Glycolytic and osmotic processes occurring during the storage of eggs caused some changes in the portions of the main parts of eggs (Table 3). Stored eggs of both feeding treatments had lowered portions of albumen, and increased portions of yolk when compared to fresh eggs (*p* < 0.05). Changes in relative portions of albumen and yolk during storage were greater in eggs of treatment E than C (*p* < 0.05). The eggshell portion was reduced in both treatments (*p* > 0.05).

Statistical analysis of external egg quality indicators (Table 4) proved that the feeding treatments and egg storage period, as well as their interaction, were not significant (*p* > 0.05) for the differences in egg weight (g) and shell strength (kg/cm^2^). The difference between feeding treatments was significant (*p* = 0.049) only for the eggshell thickness (mm).

Data presented in Table 5 refer to the influence of feeding treatments on internal egg quality indicators. The research confirmed that the feeding treatments had a significant influence on the HU and pH values of yolks (*p* = 0.01). Feeding treatments had also a highly significant influence on the albumen height and yolk color (*p* < 0.001). Storing of eggs for 28 days in a refrigerator at 4 °C proved also as highly significant for differences in albumen height, HU, yolk color, and pH values of yolk and albumen (*p* < 0.001).

Table 6 gives an overview of the amounts of selenium, vitamin E, and lutein contained in eggs. The content of lutein and vitamin E was significantly higher (*p* < 0.001) in yolks of the enriched eggs than in yolks of conventional eggs (61.45 μg/g:7.21 μg/g, and 24.03 μg/g:8.77 μg/g, respectively). Enriched eggs contained 8.94 times more lutein and 2.74 times more vitamin E than conventional eggs (Table 6). The research proved that there was 9.7 times more selenium in yolk than in albumen. Increased content of selenium in feeding mixtures resulted in increased deposition of selenium in albumen and in the yolk. Enriched eggs contained 0.114 μg selenium per g of albumen, and 0.724 μg selenium per g of yolk. Conventional eggs contained 0.063 μg selenium per g of albumen, and 0.615 μg selenium per g of yolk. Differences in the content of selenium between the treatments were highly significant (*p* < 0.001). When compared to conventional eggs content of selenium in enriched eggs was 1.81 times higher in albumen and 1.18 times higher in the yolk.

Table 7 overviews the data referring to contents of SFA, MUFA, n-6 PUFA, and n-3 PUFA in eggs (mg/100 g egg), as well as the ratio of n-6/n-3 PUFA in lipids of both groups of egg yolks. Highly significant differences were determined in the contents of myristic, pentadecanoic, and heneicosanoic fatty acid (*p* < 0.001). Differences between the treatments referring to ∑SFA in eggs were not significant (*p* = 0.513). Enriched eggs had significantly higher ∑MUFA (*p* = 0.03) than conventional eggs, with oleic fatty acid being the most represented (*p* < 0.006). Compared to the content of fatty acids in conventional eggs, enriched eggs had significantly higher ∑n-6 PUFA (*p* < 0.001). The content of n-3 PUFA was significantly more favorable in enriched eggs than in conventional eggs (ALA 265.29:99.15 mg/100 g; EPA 26.04:0.00 mg/100 g, DHA 189.32:105.43 mg/100 g; ∑n3 PUFA 480.65:204.58 mg/100 g; *p* < 0.001). The difference in ∑ n-3 PUFA between the conventional and enriched eggs was also highly significant (*p* < 0.001). The ratio n-6/n-3 PUFA was favorable in lipids of enriched egg yolks, unlike the ratio determined in conventional egg yolks of (2.19:1.00 and 8.69:1.00, respectively).

Table 8 presents the indicators of oxidation (MDA µg/g) in yolks of fresh and stored eggs. MDA values in fresh conventional eggs were significantly higher than in enriched eggs. After storing eggs for 28 days at 4 °C, MDA values in eggs of both groups were increased, yet such increase was lower in enriched eggs than in conventional eggs. Research results referring to concentrations of MDA in yolks of both groups indicated the development of oxidative process and pointed out the intensity of lipid oxidation. Compared to conventional eggs, MDA values (µg/g) in enriched eggs were lower, due to enrichment of those eggs with antioxidants, such as selenium, vitamin E, and lutein, which have protective and synergistic action. Interaction of feeding treatments and 28-day storage of eggs in refrigerator was highly significant (*p* = 0.004).

## 4. Discussion

The conducted research proved that there were biochemical processes occurring in albumen and yolk, as well as osmotic flow through the vitelline membrane from albumen to yolk and vice versa, which affected the portions of main parts in eggs. During the 28-day storage of eggs in a refrigerator at 4 °C, portions of yolk in conventional and enriched eggs were increased by 0.83% and 2.19%, respectively, and portions of albumen were lowered by 0.52% and 1.80%, respectively. Changes in portions of yolk and albumen were more pronounced in enriched eggs. As the air enters the egg through the shell pores, oxidative processes cause an increase in pH value [27,28]. Kralik et al. [4] enriched eggs with omega-3 fatty acids by adding different proportions of fish oil and microalgae. The control group contained 5% soybean oil, group E1 4% soybean oil and 0.5% fish oil + 0.5% microalgae, group E2 contained 3.5% soybean oil and 0.75% fish oil + 0.75% microalgae. Fresh yolks had a pH value ranging from 6.05 in the control group to 6.12 in the E2 group, and the pH of stored eggs increased from 6.31 in the control group to 6.36 in the E1 group. In our research, eggs stored in a refrigerator had increased pH in yolks and albumen of both groups, yet such an increase in pH was significantly higher in enriched eggs. It is known that ovomucin represents about 3.5% of the total egg albumen protein and is responsible for the formation of the thick gel-like part of albumen. Ovomucin is insoluble at neutral pH. The pH values in fresh egg albumen range from 8.3 to 8.6, and during storage, they increase up to 9.7, depending on the storage duration. An increase in the albumen pH causes disulfide bonds to split and release carbohydrate chains, which results in liquid albumen. It loses height, and consequently, the HU also decreases [29,30]. In our paper, HU values in fresh eggs are higher in the E group compared to the C group. During the egg storage, HU decreases, which is also the case in our experiment, however, the decrease in HU values is less pronounced in group E than in group C. We assume that this is the effect of added nutricines that have an antioxidant effect. Similar observations are made by Gajčević et al. [12], who state that HU values in eggs from laying hens fed 0.4 mg/kg of organic selenium are more stable during egg storage compared to eggs from laying hens fed mixtures with a lower level of organic selenium. Furthermore, the influence of feeding treatment on eggshell thickness was determined (*p* = 0.049). In enriched eggs, slightly lower values for eggshell thickness were measured compared to conventional eggs, which we assume is an effect of lutein’s inhibition of the estrogenic activity of several tissues. The impact is particularly visible in fresh enriched eggs, where the value of shell thickness is significantly lower compared to stored conventional eggs.. Similar observations are made by de Oliveira et al. [31]. The color of the yolk depends on the concentration of lutein in the feed and on the storage period (*p* < 0.01), as well as on their interaction (*p* = 0.003). Lutein cannot be synthesized in human or animal organisms, so it has to be taken in through food. Egg yolk is a matrix composed of digestible lipids, cholesterol, triacylglycerols, and phospholipids, and the carotenoids lutein and zeaxanthin are dispersed in this matrix along with other fat-soluble micronutrients [32]. Yolk color varies from light to dark orange depending on the deposition of xanthophylls from the feed. Manufacturers add natural and artificial pigments to animal feed in order to enhance the yolk color. In various studies, scientists modified the content of pigments in laying hens’ feed in order to enhance the yolk color. Consumers prefer eggs with yolks of a certain color, and they are informed about it. Yolk color can be measured with a color scale (Yolk Color Fan (YCF)) or a colorimeter where the color is expressed through three values CIE L* (degree of lightness), CIE a* (degree of redness) and CIE b* (degree of yellowness). If the color scale (YCF) is used to measure the color of the yolk, European consumers prefer coloring between 9 and 14. In Europe, there is a difference in consumer preferences for yolk color between northern and southern countries. Consumers from Southern European countries prefer intensely colored yolks (11–14), while those from Northern European countries prefer paler yolks (9–10) [33] (Grashorn, 2016). In a survey conducted by Berkhoff [34], respondents stated that egg yolks from laying hens kept in alternative housing systems were yellower than those produced on a farm. However, when buying eggs, they still choose eggs produced in cage systems, as their price is lower. For this reason, egg producers shall design feeding mixtures for laying hens to increase the intensity of yolk color. In the study into the influence of carotenoids supplemented with laying hens’ feed on the yolk color intensity, Kojima et al. [35] reported that supplementation of 60 mg/kg lutein available as commercial extract of calendula flowers had a significant effect (*p* ˂ 0.001) on yolk color. Fletcher and Halloran [36] state that the addition of a higher concentration of lutein from marigold extract (60 mg/kg) to the feed of laying hens affects the yolk color intensity, which was darker in the experimental group compared to the control group (CIE L* = 55, 68 or CIE L* = 59.92). The results obtained in our research are in accordance with theirs. Hens have the ability to utilize from 20 to 60% of pigments from feed into the yolk [37]. We think that the higher content of lutein in the feed of the experimental group of hens influenced the more intense yolk color compared to the control group, both in fresh (13.30 and 11.58) and stored eggs (13.25 and 11.05).

Steinberg [38] reported that conventional eggs contained 0.3 to 0.5 mg of total xanthophyll, of which 50% was lutein. Nolan et al. [39] stated that the bioavailability of lutein from food depended on whether lutein was available in the free or esterified form. Marigold (*Tagettes erecta*, L.) contains significant amounts of xanthophylls, mainly lutein and zeaxanthin, which are used for the coloring of egg yolks [40]. Golzar Adabi et al. [41] supplemented feeding mixtures based on corn and soybean with 0, 250, 500, and 750 ppm lutein by using commercial preparation FloraGlo as a dietary supplement. The greatest enrichment of eggs with lutein occurred with supplementation of 750 ppm (1.43 mg/57 g egg), however, supplementation with already 250 ppm influenced a significant increase in lutein in eggs when compared to the control group (1.35 mg/57 g egg, and 1.12 mg/57 g egg, respectively; *p* ˂ 0.01). At higher amounts of lutein supplemented to feed, deposition of lutein in the egg was slowed. Nain [42] fed laying hens with diets that contained 10% linseed oil and 500 mg/kg lutein over 56 days. The highest deposition of lutein was determined in the experimental group after 28 days, being 34.59 µg/g yolk, compared to the control 8.64 µg/g yolk. Skřivan et al. [43] reported that supplementation of 950 mg/kg marigold flower extract to laying hens’ diet could influence the increase in lutein concentration from 12.3 to 36.3 mg/g yolk. Grčević et al. [44] enriched omega-3 eggs with lutein. They added 0.1% and 0.2% marigold flower extract to the laying hens’ feed. The lutein content in egg yolks was analyzed on the 21st and 35th day of the feeding treatment and amounted to 20.11 μg/g and 22.57 μg/g in the control group, 107.41 μg/g and 103.74 μg/g in the O1 group, and 118.86 μg/g and 113.31 μg/g in O2 group. Islam et al. [45] reported faster deposition of lutein in yolk when laying hens were fed a diet with low concentrations of lutein. In their research focusing on the enrichment of table eggs with functional ingredients, Kralik et al. [10] pointed out that supplementation of laying hens’ diet with 200 mg/kg lutein resulted in a significant increase in lutein in yolks of the experimental group when compared to the control (104.95 μg/g and 12.44 μg/g, respectively; *p* ˂ 0.001). If the data of the mentioned authors on the use of MFE as a source of lutein are compared with the results of this paper, it can be assumed that after intake and transport of lutein through the digestive system, lutein is efficiently deposited in egg yolks. The lutein content in eggs depends on the source and the amount of lutein that is ingested through feed.

Skřivan et al. [46] and Tufarelli et al. [47] reported protective effect of lutein and selenium against oxidation in yolks, respectively. Jiang et al. [48] determined the protective role of high concentrations of vitamin E contained in n-3 PUFA enriched eggs (200 mg/kg feed). The research of Chen et al. [49] pointed out that α-tocopherol had antioxidative effect in concentrations below 50 μg/g yolk and pro-oxidative effect if its concentration in yolk was above 75 μg/g. In the presented research, the content of total vitamin E (tocopherol) was 8.77 μg/g of yolk in the control, and 24.03 μg/g of yolk in the experimental group, which was below the stated pro-oxidative concentration. Surai et al. [50] and Ren et al. [51] confirmed the protective action of vitamin E in n-3 PUFA-enriched eggs. Supplementation of feeding mixtures with higher amounts (200 mg/kg) of vitamin E increased the antioxidative effect and reduced the concentration of MDA in yolks. Those results are in line with ours, as this research also proved reduced oxidative changes occurring in yolks of group E. Selenium is an essential part of the selenoenzyme GPH-Px. It is present in egg albumen and yolk and protects cells from lipid oxidation [52,53,54]. Fašiangova et al. [20] reported that selenium reduced the oxidation processes and prevented an increase in pH value. When feeding laying hens diets supplemented with 0.46 mg/kg organic selenium and 100 mg vitamin E, Gjorgovska and Filev [55] reported that 100 g of yolk contained 21.14 µg/g selenium. The concentration of MDA is observed as an indicator of lipid oxidation which occurs during egg storage [10,28]. Liang et al. [56] reported that the TBARS values increased during the period of storing eggs at 4 °C (from 0 to 24th day), both in conventional and in n-3 PUFA enriched eggs. Such results are in accordance with results obtained in this research, as well as with results published by Mohiti-Asli et al. [57], who pointed out that storing eggs at low temperatures had less influence on egg lipid oxidation. Wang et al. [36] added 0.3 mg/kg organic selenium, which affected the lowering of MDA concentration in fresh and stored eggs. As of the research of Kralik et al. [28], the concentration of MDA in fresh conventional eggs was 0.597 µg/g, and in eggs stored for 28 days in a refrigerator at 4 °C it was 0.709 µg/g. The same authors reported the lipid oxidation in omega-3 eggs was 0.510 µg/g, and after 28-day long storage in a refrigerator at 4 °C, it was 0.657 µg/g. Results reported by the stated authors are slightly lower, however, the trend in values referring to the oxidation in control and enriched eggs correspond to our results. The analysis of the results of our research and that of other authors on the enrichment of eggs with selenium, vitamin E, and lutein shows that eggs can be successfully enriched with the mentioned nutrients at their increased concentrations in feed. The antioxidant effect of added amounts of selenium, vitamin E, and lutein was determined by analyzing the concentration of MDA in enriched eggs compared to conventional eggs.

Commercial table eggs contain higher portions of n-6 PUFA (mostly C18.2 n-6, linoleic, LA, USA), yet they are deficient in n-3 PUFA. Studies have been performed to enrich eggs with n-3 PUFA in order to reduce the ratio n-6/n-3 PUFA to the recommended 4-5:1 [58]. Omega 3 fatty acids originate from plant oils and marine oils [59]. Our research confirmed that enriched eggs contained 2.7 times more ALA, 26.0 times more EPA and 1.8 times more DHA than conventional eggs. The ratio n-6/n-3 PUFA in enriched egg yolk lipids was considerably more favorable than the same ratio of conventional eggs (2.19:1 and 8.69:1, respectively). Zdunczyk et al. [60] pointed out the synergistic action of antioxidants in preventing the oxidation of long-chain fatty acids (LC- PUFA), which was also the case in our research. ALA is a precursor of n-3 PUFA, and conversion into EPA and DHA amounts to 5%, depending on the content of n-6 PUFA in feed [61,62]. Enriched eggs are desirable in human nutrition because they contain bioactive substances, n-3 PUFA, selenium, vitamin E, and lutein, as well as the recommended ratio of n-6/n-3 PUFA [63]. Abedi and Sahari [64] reported that standard eggs contained 0.1% EPA, 0.7% DHA, and 0.8% ALA in the yolk. Kralik et al. [6] were supplementing feeding mixtures with different amounts of fish oil and managed to increase the concentrations of EPA and DHA. The content of DHA in egg yolks also increased, but not linearly because it depended on the LA/ALA ratio, which affected the synthesis of n-3 PUFA. There is also a competition occurring in synthesis, as there are the same enzymes, i.e., Δ5 and Δ6 desaturases required. Irawan et al. [65] determined a linear relationship between the content of ALA in feed and the content of EPA and DHA in total n-3 PUFA in egg yolks, with a simultaneous decrease in the content of LA. Authors reported that the addition of 100 g ALA/kg feed resulted in 126 mg/egg DHA. In order to increase n-3 fatty acids in eggs, Omri et al. [66] fed hens mixtures with the addition of flaxseed meal and vegetable powder (tomatoes and peppers). The results of their research indicate that the increase in LA in feed affected the LA/ALA ratio, and the content of EPA and DHA in egg yolks decreased. It was also confirmed that lowering LA and increasing ALA content in poultry feed influenced the synthesis of DHA. Feng et al. [11] (2020) state that adding more than 0.91% of microalgae oil or more than 3.38% of fish oil can produce an egg with at least 200 mg of DHA. By consuming one such egg, the daily needs of infants and pregnant and nursing women would be almost met, while adults would meet about 80% of the daily needs for DHA. The results of our research indicate that an adult person consuming 100 g of enriched eggs can meet approximately 86% of the daily needs for EPA and DHA; for ALA approximately 16% for men and approximately 24% for women; approximately 52% for selenium, approximately 4.5% for vitamin E, and approximately 17% for lutein. The results of our research show that eggs enriched with more nutrients simultaneously represent a quality source of these nutrients in increased concentrations in order to meet nutritional needs.

## 5. Conclusions

The study presented by this paper focused on fish oil that contains EPA and DHA, as well as rapeseed and linseed oil, which are rich in ALA. Compared with group C, those oils supplemented with group E laying hens’ diet influenced better deposition of ALA and DHA in the egg yolks, which increased total n-3 PUFA and decreased n-6 PUFA, especially arachidonic fatty acid. Combinations of oils supplemented with laying hens’ feed resulted in eggs enriched with a significant amount of n-3 PUFA, and with selenium, vitamin E, and lutein that facilitate the protective, synergistic effect of antioxidative ingredients in eggs. When consuming such eggs enriched with several functional ingredients, people are sure to take in essential components that are important for their health.

## Figures and Tables

**Table 1 animals-13-00321-t001:** Composition and chemical analysis of basic feeding mixture.

Ingredients	%	Chemical Analysis **	%
Corn	48.47	Moisture	9.30
Soybean cake	22.33	Crude protein	16.63
Toasted soybean	3.00	Crude fat	7.30
Sunflower cake	5.00	Crude fibre	4.00
Alfalfa	1.67	Ash	16.54
Calcium granules	10.33		
Monocalcium phosphate	1.33		
Yeast	0.50		
Salt	0.33		
Acidifier	0.33		
Minerals nanofeed	0.33		
Methionine	0.15		
Premix *	1.20		
Soybean oil	5.00		
Total	100.00	ME, MJ/kg	11.60

* Premix (1 kg) contains: vitamin A 834000 IU, vitamin D3 208500 IU, vitamin K3 167 mg, vitamin B1 150 mg, vitamin B2 374 mg, vitamin B6 200 mg, vitamin B12 918 μg, vitamin C 1860 mg, niacin 2085 mg, pantothenic acid 584 mg, folic acid 75 mg, biotin 7 mg, choline chloride 33,600 mg, iron 2520 mg, iodine 76 mg, copper 425 mg, manganese 5640 mg, zinc 5175 mg, canthaxanthin 300 mg. ** Referential methods applied in chemical analysis of feeding mixtures: moisture HRN ISO 6496; ash HRN EN ISO 5984; crude protein HRN ISO 5983-2; fat HRN; crudefibere HRN EN ISO 6865, modified (Croatian standards, 2001; 2004; 2010).

**Table 2 animals-13-00321-t002:** The content of fatty acids in feeding mixtures (% in the sum of fatty acids).

Fatty Acid	Treatment C	Treatment E
Myristic (C14:0)	0.17 ± 0.01	1.54 ± 0.01
Pentadecanoic (C15:0)	0.05 ± 0.01	0.25 ± 0.01
Palmitic (C16:0)	14.69 ± 0.02	12.12 ± 0.01
Heptadecanoic (C 17:0)	0.13 ± 0.01	0.26 ± 0.01
Stearic (C 18:0)	5.52 ± 0.02	3.60 ± 0.01
Arachidic (C 20:0)	0.46 ± 0.01	0.38 ± 0.01
Behenic (C 22:0)	0.21 ± 0.01	0.25 ± 0.01
∑ SFA	21.23 ± 0.24	18.41 ± 0.01
Palmitoleic (C 16:1)	0.23 ± 0.01	1.66 ± 0.01
Oleic (C18:1 cis + trans)	28.27 ± 0.01	33.13 ± 0.02
Eicosenoic (C 20:1)	0.17 ± 0.01	1.07 ± 0.01
Erucic (C 22:1)	1.16 ± 0.01	0.73 ± 0.02
∑ MUFA	29.85 ± 0.05	36.61 ± 0.01
Linoleic (C18:2 n-6)	44.17 ± 0.01	23.13 ± 0.03
∑ PUFA n-6	44.17 ± 0.01	23.13 ± 0.03
Alfa linolenic (C18:3 n3)	4.24 ± 0.01	17.39 ± 0.02
Eicosapentaenoic (C20:5 n-3)	0.49 ± 0.00	1.62 ± 0.03
Docosahexaenoic (C 22:6 n-3)	0.00 ± 0.00	2.81 ± 0.02
∑ PUFA n-3	4.73 ± 0.01	21.83 ± 0.07
∑ PUFA n-6/ PUFA n-3	9.32 ± 0.01	1.06 ± 0.01

SFA = saturated fatty acids; MUFA = monounsaturated fatty acids; PUFA = polyunsaturated fatty acids; average value of the results of two sample analyses is shown; C = 5% soybean oil, 0.32 mg/kg organic selenium, 25.20 mg/kg vitamin E, and 20.5 mg/kg lutein; E = 1.5% fish oil + 1.5% rapeseed oil + 2.0% linseed oil, 0.47 mg/kg organic selenium, 125.20 mg/kg vitamin E, and 120.50 mg/kg lutein.

**Table 3 animals-13-00321-t003:** Influence of feeding treatment and storage period on the portions of main parts in eggs.

Treatment/ Group	Time of Analysis	Albumen Portion (%)	Yolk Portion (%)	Shell Portion (%)
Conventional	Fresh	60.47 ^b^	26.51 ^a^	13.02
	Stored	59.94 ^b^	27.34 ^a^	12.72
Enriched	Fresh	62.04 ^a^	25.21 ^b^	12.75
	Stored	60.25 ^b^	27.39 ^a^	12.36
SEM		0.485	0.463	0.262
*p* value	Feeding treatment	0.058	0.176	0.257
	Storage period	0.019	0.001	0.184
	Interaction	0.197	0.147	0.860

^a,b^ Letters in superscript represent the difference between values shown in the column; C = 5% soybean oil, 0.32 mg/kg organic selenium, 25.20 mg/kg vitamin E, and 20.5 mg/kg lutein; E = 1.5% fish oil + 1.5% rapeseed oil + 2.0% linseed oil, 0.47 mg/kg organic selenium, 125.20 mg/kg vitamin E, and 120.50 mg/kg lutein.

**Table 4 animals-13-00321-t004:** Influence of feeding treatments on external egg quality indicators.

Treatment/ Group	Time of Analysis	Egg Weight (g)	Shell Thickness (mm)	Shell Strength (kg/cm^2^)
Conventional	Fresh	63.79	0.421 ^ab^	2.867
	Stored	65.98	0.423 ^a^	2.937
Enriched	Fresh	66.29	0.404 ^b^	2.841
	Stored	65.38	0.415 ^ab^	2.759
SEM		0.987	0.006	0.145
*p* value	Feeding treatment	0.340	0.049	0.485
	Storage period	0.520	0.287	0.966
	Interaction	0.119	0.471	0.601

^a,b^ Letters in superscript represent the difference between values shown in the column; C = 5% soybean oil, 0.32 mg/kg organic selenium, 25.20 mg/kg vitamin E, and 20.5 mg/kg lutein; E = 1.5% fish oil + 1.5% rapeseed oil + 2.0% linseed oil, 0.47 mg/kg organic selenium, 125.20 mg/kg vitamin E, and 120.50 mg/kg lutein.

**Table 5 animals-13-00321-t005:** Influence of feeding treatments on internal egg quality indicators.

Treatment/ Group	Time of Analysis	Albumen Height (mm)	HU	Yolk Color	Albumen pH	Yolk pH
Conventional	Fresh	6.01 ^b^	74.29 ^b^	11.58 ^b^	8.51 ^b^	5.81 ^b^
	Stored	4.56 ^c^	61.59 ^c^	11.05 ^c^	8.80 ^a^	6.18 ^a^
Enriched	Fresh	6.90 ^a^	80.74 ^a^	13.30 ^a^	8.43 ^b^	5.74 ^c^
	Stored	4.97 ^c^	65.47 ^c^	13.25 ^a^	8.78 ^a^	6.13 ^a^
SEM		0.227	1.968	0.124	0.033	0.023
*p* value	Feeding treatment	˂0.001	0.01	˂0.001	0.133	0.01
	Storage period	˂0.001	˂0.001	˂0.001	˂0.001	˂0.001
	Interaction	0.289	0.515	0.003	0.306	0.822

^a,b,c^ Letters in superscript represent the difference between values shown in the column; C = 5% soybean oil, 0.32 mg/kg organic selenium, 25.20 mg/kg vitamin E, and 20.5 mg/kg lutein; E = 1.5% fish oil + 1.5% rapeseed oil + 2.0% linseed oil, 0.47 mg/kg organic selenium, 125.20 mg/kg vitamin E, and 120.50 mg/kg lutein.

**Table 6 animals-13-00321-t006:** Content of selenium, vitamin E, and lutein in eggs.

Ingredient	Conventional	Enriched	*p* Value
Selenium in albumen (μg/g)	0.063 ^b^	0.114 ^a^	˂0.001
Selenium in yolk (μg/g)	0.615 ^b^	0.724 ^a^	˂0.001
Lutein in yolk (μg/g)	7.21 ^b^	61.45 ^a^	˂0.001
Vitamin E in yolk (μg/g)	8.77 ^b^	24.03 ^a^	˂0.001

^a,b^ Letters in superscript represent the difference between values shown in the row *p* < 0.05; C = 5% soybean oil, 0.32 mg/kg organic selenium, 25.20 mg/kg vitamin E, and 20.5 mg/kg lutein; E = 1.5% fish oil + 1.5% rapeseed oil + 2.0% linseed oil, 0.47 mg/kg organic selenium, 125.20 mg/kg vitamin E, and 120.50 mg/kg lutein.

**Table 7 animals-13-00321-t007:** Fatty acid profile in egg yolks (mg FA /100 g of eggs).

Fatty Acid	Conventional	Enriched	*p* Value
Myristic (C14:0)	20.19 ± 2.28 ^b^	27.11 ± 1.43 ^a^	˂0.001
Pentadecanoic (C15:0)	4.18 ± 1.23 ^b^	7.46 ± 0.37 ^a^	˂0.001
Palmitic (C16:0)	1585.93 ± 343.8	1511.75 ± 198.5	0.687
Heptadecanoic (C17:0)	15.47 ± 4.30	16.27 ± 1.83	0.714
Stearic (C18:0)	564.05 ± 146.70	469.02 ± 63.40	0.220
Heneicosanoic (C21:0)	16.84 ± 2.10 ^a^	5.63 ± 0.41 ^b^	˂0.001
∑SFA	2206.66 ± 489.7	2037.24 ± 258.2	0.513
Palmitoleic (C16:1)	141.53 ± 27.0 ^b^	241.74 ± 18.42 ^a^	˂0.001
Heptadecenoic (C17:1)	11.50 ± 4.49 ^b^	18.96 ± 0.93 ^a^	0.006
Oleic (C18:1)	2616.81 ± 250.1 ^b^	3124.11 ± 177.2 ^a^	0.006
Eicosenoic (20:1)	12.37 ± 0.80	13.87 ± 1.22	0.051
∑MUFA	2782.21 ± 267.9 ^b^	3398.69 ± 194.2 ^a^	0.003
Linoleic (C18:2 n-6)	1637.96 ± 194.3 ^a^	985.98 ± 61.0 ^b^	˂0.001
Eicosadienoic (C20:2 n-6)	10.02 ± 1.2 ^b^	6.92 ± 1.2 ^b^	0.004
Arachidonic (C20:4 n-6)	130.51 ± 16.5 ^a^	62.45 ± 4.9 ^b^	˂0.001
∑n-6 PUFA	1778.49 ± 208.8 ^a^	1055.35 ± 66.5 ^b^	˂0.001
α-linolenic (C18:3 n-3)	99.15 ± 14.8 ^b^	265.29 ± 21.9 ^a^	˂0.001
EPA (C20:5 n-3)	0.00 ± 0.00 ^b^	26.04 ± 2.3 ^a^	˂0.001
DHA (C22:6 n-3)	105.43 ± 15.5 ^b^	189.32 ± 16.3 ^a^	˂0.001
∑n3 PUFA	204.58 ± 28.2 ^b^	480.65 ± 35.9 ^a^	˂0.001
n-6/n-3 PUFA	8.69 ± 0.4 ^a^	2.19 ± 0.1 ^b^	˂0.001

^a,b^ Letters in superscript represent the difference between values shown in the rows *p* < 0.05. SFA = saturated fatty acids; MUFA = monounsaturated fatty acids; PUFA = polyunsaturated fatty acids. C = 5% soybean oil, 0.32 mg/kg organic selenium, 25.20 mg/kg vitamin E, and 20.5 mg/kg lutein; E = 1.5% fish oil + 1.5% rapeseed oil + 2.0% linseed oil, 0.47 mg/kg organic selenium, 125.20 mg/kg vitamin E, and 120.50 mg/kg lutein.

**Table 8 animals-13-00321-t008:** Lipid oxidation in egg yolks.

Experimental Groups	Time of Analysis	µg MDA/g of Yolk
Conventional	Fresh	0.939 ^b^
	Stored	1.085 ^a^
Enriched	Fresh	0.892 ^c^
	Stored	1.016 ^ab^
SEM		0.040
*p* value		
Feeding treatments		0.168
Storage period		0.792
Interaction		0.004

^a,b,c^ Letters in superscript represent the difference between values shown in the column; C = 5% soybean oil, 0.30 mg/kg organic selenium, 25 mg/kg vitamin E, and 20.5 mg/kg lutein; E = 1.5% fish oil + 1.5% rapeseed oil + 2.0% linseed oil, 0.45 mg/kg organic selenium, 125.20 mg/kg vitamin E, and 120.50 mg/kg lutein.

## Data Availability

The data analyzed for the study are available from the corresponding author upon reasonable request.

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
