# Peer review of "Natural Multi-Enriched Eggs with n-3 Polyunsaturated Fatty Acids, Selenium, Vitamin E, and Lutein"

_animals, 2023, doi:10.3390/ani13020321_

Round 1

Reviewer 1 Report

The paper entitled Natural multi-enriched eggs is interesting, and presents satisfactory results, however, some major revision is required before further processing.

Introduction

Obs 1. The title does not present sufficient information. Enriched in what and how.

Obs 2. In the abstract. Portions of n-3… I will suggest to mention exactly with what n-3 fatty acids the eggs were enriched.

Obs 3. Row 53. Please delete forages and use ingredients, forages are mostly used in ruminants’ nutrition, not for poultry. Also please find a reference to support that statement. I can recommend the following https://doi.org/10.1038/s41598-021-00343-1

Obs 4. Row s 54-55. The sentence must be rephrased. It is not accurate is incomplete and incoherent. The vegetable by-products must be mentioned also. For example, as it is very well known, fish oil, fish meal, flax seed, oil and meal, pumpkin meal, rapeseed, algae and some oils are the most studied feed ingredients for monogastric, especially poultry, to enrich the eggs in n-3 PUFA. Also, add some background studies where eggs enriched have been obtained (with or without antioxidants, carotenoids and minerals).

The aim of the paper is hot very clear and the hypothesis is missing. Eggs enriched in PUFA, antioxidants and minerals is not something new, the authors must add a sentence explaining what is the novelty of the current study.

Material and Methods

For experimental diet. If the authors replaced the Soybean oil entirely, the experimental diet must be given in Table 1. Also, the content of selenium, vitamin E and lutein must be given for experimental diet versus control diet.

The vitamin E supplement in the experimental diet, was added over the 2085 mg/kg vitamin E from premix? That means that experimental diet was supplemented with 2210.2 mg vitamin E/kg.

The production performances are not given in this current paper, which I think the authors have published them elsewhere. I would like to know if the laying intensity was 100% in each group.

Also, it is interesting to know how the eggs were collected? 50 hens per group means 50 eggs per day (if the laying was 100%). The authors mentioned that 100 eggs were analysed. How these eggs were collected? In the last experimental day?

Stored eggs at 4C for 28 days are already achieved, although not many papers studied the shelf life of the eggs during storage at different temperatures under the effect of fat diets.

Why the authors analysed only 10 egg yolks for FA, lipid oxidation and lutein determinations?

The no of samples for selenium analysis is wrong. The authors collected 40 eggs. From these eggs they should have 40 yolks and 40 albumens from which 20 yolks and albumens in C and 20 yolks and albumens in E group.  Please revise.

Why the vitamin E was determined only from 5 yolks/group? The no of observation is too low and the results is not accurate at all to run a statistical analysis.

Results:

If the FA analyses for compound feed were determined in triplicate, the values should be given with STD.

All the results, must be moved above the Tables, not under. Please correct.

Table 3 is confusing. The studied supplements (Se, vit E and lutein) were supplemented in the compound feed. In table 3, the results of the chemical composition are given. Is not very clear how much was supplemented in the diet on E group.

These part needs consistent clarification.

Figure 1, is wrong presented. The fresh eggs of C and E groups must be placed next to each other. Same for the stored eggs. Use black font in the figure notation.  In this form the figure is very confusing and it looks like a chop soup of bars and values.

For the results presented in table 4 the authors used two-way anova, in the material and methods this is not mentioned. Same observation for table 5 and 8.

Discussions

Row 271. It is not long storage. Is a normal period of time. Over 30 days of storage is long period. Generally, commercial eggs have 30 days of shelf life. Rephrase it.

Why the HU were higher in the e group, versus the C group. Is there any link between the supplements on this parameter? Can the lutein act as an antioxidant also to prevent the Hu to decrease, or the implications of vitamin E? please explain this. Some responses can be found in literature, regarding the effect of antioxidants on shelf life of the eggs.

Row 303 – 333. In this part the authors made a review of the literature. There is a lack of own interpretation of the results from this study, corroborated with literature data. Please improve this part.

Row 334 – 374. Please shorten this part and include some of your own ideas. Same observation as above.

Reference 47 is about lipid me6tabolism in meat, not in eggs. Please replace it with a reference that deals with this subject, for example https://doi.org/10.3390/antiox11101948 , https://doi.org/10.1007/s11745-001-0792-7 , https://doi.org/10.1093/ps/79.7.971

In the discussion part is lacking on explaining what was the outcome on the use of such supplements. The authors have not evidenced concrete what was the effect of selenium, vitamin E or lutein, beside the fact that the content of these nutrients increased. Also, the synergistic or antagonistic effects of these nutrients on poultry diets is not explained.

Further, what is the transfer rate of this nutrients from feed to food and to consumers? 

In short sentences some responses must be given and the paper need to be “enriched” with additional information. In this form, this paper do not bring much of novelty.

The second and the most important problem is the experimental design. Low no of observation for some important parameters …

 Best of luck!

Author Response

Dear Reviewer, please find our answers to your remarks in attached document.

Best regards

Reviewer 2 Report

The objective of the study is to investigate the quality and shelf life of eggs from birds receiving enriched diets. The topic is current and needs further research. The work is interesting, and it is well written, but it needs a little revision before publication.

Line 50 - Include reference

Introduction: Please improve the state-of-the-art analysis to show the progress beyond the state of the art clearly. This would provide the readers with a sense of continuity and help them place your paper in a better context, very much strengthening your article's impact.

Is there any legislation to classify eggs as enriched? Mineral levels? Are there any requirements, conditions, attributes or any period analysis that must be performed to prove these characteristics?

Is there any relationship with the daily needs of human beings?

Is this rule universal or can it be modified in different countries or continents?

Is information available on the amount of enriched eggs currently produced? And information on production costs and the final price of the product for consumers?

The objective needs to be written more clearly.

Replace Treatment C and Treatment E with conventional eggs and enriched eggs throughout the text.

Replace Treatment C and Treatment E with conventional eggs and enriched eggs in the tables.

Why weren't the eggs stored at room temperature? This would make it easier to observe the loss of quality. 28 days of storage in refrigerated eggs is a short period.

Talk about the effects of refrigeration on stored eggs. Include the following reference https://doi.org/10.1080/00439339.2022.2105276

Line 402 - The sentence needs to be rewritten.

Line 409 - 411 - Rewrite sentence.

What are the market trends for enriched eggs?

What are the search trends on the topic covered?

Author Response

Dear Reviewer, please find answers to your remarks in attached document.

Best regards

Round 2

Reviewer 1 Report

The introduction part was improved; however, I am not very satisfied on how it is arranged.

Row 69 – delete P5

Row 98-99. That sentence should sound like “ To our knowledge there are little information in the literature regarding the simultaneous enrichment with …. of the eggs” or something like that. As I have indicated previously, there are some studies dealing with similar subjects.

Row 108 -109. Add the age of the hens.

After the diets, add the provenience of the supplements used (selenium, vitamin E and lutein). How these nutrients have been mixed through the diet to assure the uniformity?

Row 147-148. Regarding this observation, now I have understood how they were collected and analysed, however, is wrong explained. The sentence should be reversed. First mention the no of eggs, and after how many samples were formed. For example:  The content of selenium in eggs was determined on 20 eggs/group from which we formed 10 yolk and 10 albumen samples.  

This is for all explanations about samples preparation.

Table 3 is useless. This information is already presented in Table 1. With other words, the authors says that they supplemented the diets with 0.47 mg/kg selenium, 125.20 mg/kg vitamin E and 120.50 mg/kg kg of lutein, and the exact same amounts were determined by the chemical composition. This is very wrong. Corn was the main ingredient of the diet, so corn in the main source of xantophills, especially for poultry. Also, in the diets alfalfa was added, which is also a natural source of lutein…

Please see this reference Perry, A., Rasmussen, H., & Johnson, E. J. (2009). Xanthophyll (lutein, zeaxanthin) content in fruits, vegetables and corn and egg products. Journal of food Composition and Analysis, 22(1), 9-15.

In figure 1, the values do not account 100%. It will be better if presented as a table as the rest of the parameters.

General comment. Please use the expression significantly higher, instead of statistically significantly higher. It does not sound very scientific.

Row 346-348. From where the authors obtained those values? What E1 and E2 groups?

Row 368-386. Regarding the yolk color, the only statement that the authors have is that Results obtained in this research are in accordance with theirs.

General comment. For the entire Results chapter, please, do not repeat the values from the tables in the text, this can be done only in the discussion section, when comparing with other studies.

Row 387-413. This part is a literature review. Due to the fact that authors mostly confirmed previously published papers, shows the lack of own data interpretation.

Rows 418 – 488. Was improved adequately.

Row 497-498. Please delete or rephrase this part. Eggs are not eaten row, and by cooking them, some of the nutrients are lowered due to high cooking temperature.

Please be careful at the number of self-citations.

Author Response

Dear Reviewer,

please find replies to Your comments in attached document. All changes made in text of the manuscript are highlighted in green color.

Best regards
